# Proof of Equivalence of Carnot Principle to II Law of Thermodynamics and Non-Equivalence to Clausius I and Kelvin Principles

**DOI:** 10.3390/e24030392

**Published:** 2022-03-11

**Authors:** Grzegorz Marcin Koczan

**Affiliations:** Department of Mechanical Processing of Wood, Institute of Wood Sciences and Furniture, Warsaw University of Life Sciences, 02-776 Warsaw, Poland; grzegorz_koczan@sggw.edu.pl

**Keywords:** II law of thermodynamics, Carnot principle, Kelvin principle, Ostwald principle, *perpetuum mobile* type III, Clausius I and II principles, formal implication, model theory

## Abstract

The II law of thermodynamics is most often given in three supposedly equivalent formulations: two Clausius (I and II) and one Kelvin. The most general and indisputable entropy formulation belongs to Clausius (II). The earlier Clausius I principle determines the natural direction heat flow between bodies at different temperatures. On the other hand, the Kelvin principle states that it is impossible to completely convert heat into work. The author argues that the Kelvin principle is a weaker statement (or more strictly non-equivalent) than the Clausius I principle, and the latter is a weaker statement than Carnot principle, which is equivalent to Clausius II principle. As a result, the Kelvin principle and the Clausius I principle are not exhaustive formulations of the II law of thermodynamics. At the same time, it turns out that the Carnot principle becomes such a formulation. Apart from providing a complete set of proof and disproof, the author, indicates where the methodological errors were made in the alleged proof of the equivalence of the Kelvin principle and both Clausius principles.

## 1. Introduction

We shall start with the chronological formulation of the fundamental principles of thermodynamics according to Carnot, Clausius I, Kelvin and Clausius II. These four principles are closely related to the II law of thermodynamics. It can be said that the Carnot principle opened the way to the formulation of the II law of thermodynamics, and the Clausius II principle fully achieved this goal. It is commonly claimed (allegedly proven) that the principles of Clausius I and II and the Kelvin principles are equivalent, and thus they constitute the II law of thermodynamics. On the other hand, the Carnot principle is claimed to be only a consequence of the II law of thermodynamics, not its equivalent formulation. Therefore, the relationship between all four principles needs to be verified.

In order for the formulated principles to be exact, it is necessary to define the context: the virtual space of thermodynamic processes and the elementary rules by which they are governed. The processes will be represented on the basis of diagrams characteristic of heat engines or heat pumps (see Figure 1). Such diagrams include a heat reservoir (thermostat) with a temperature of $T_{1}$ (heater) and a heat reservoir (thermostat) with a temperature of $T_{2}$ (cooler, $T_{2}<T_{1}$). The reservoirs can exchange heat naturally between each other or with a device that can perform work (engine) or, at the expense of work, force unnatural heat transfer between the reservoirs (heat pump or refrigerator). The diagrams can also show impossible virtual thermodynamic processes. The role of the formulated thermodynamic principles is to narrow the set of thermodynamic processes to a set of physically possible processes. Nevertheless, the initial Ω set of all considered diagrams (processes) must also include non-physical processes. Otherwise, it would not be possible to prove or disprove the implications of the two principles, which are often based on the negation of physical processes. The set of diagrams of Ω virtual processes will, however, be limited by the assumption that the I law of thermodynamics (or more precisely, the law of energy conservation) is met ($Q_{1}=W+Q_{2}$).

The simplicity of the diagrams allows them to be added naturally, reflecting the superposition of physical processes. Such a superposition appears implicitly in any textbook proof of the equivalence of the thermodynamic principles which are under consideration in this work. It is, therefore, natural that where two diagrams comply with a given principle, their sum should also comply with that principle. In other words, it is assumed that the set of diagrams in accordance with a given physical principle should be a convex set in the above-mentioned sense.**Carnot principle (C0)***The efficiency of the heat conversion process $Q_{in}>0$ to work W in the device operating in the range between the temperature of the heat source $T_{in}$ and the temperature of the heat receiver $T_{out}$, cannot be greater than the ratio of the difference of these temperatures to the temperature of the heat source:*(1)$$\frac{W}{Q_{in}}\leq \frac{T_{in}-T_{out}}{T_{in}}.$$Such a formulation does not explicitly contain the information that the temperature of the heat source must be higher than that of the heat receiver. However, the implicit condition that the engine performs real positive work (W>0) entails the aforementioned temperatures relationship $T_{in}=T_{1}>T_{out}=T_{2}$. In this case, the condition of the Carnot principle takes the form:(2)$$\eta :=\frac{W}{Q_{1}}\leq \frac{T_{1}-T_{2}}{T_{1}}=:\eta_{C}.$$

Mathematically speaking, we can also consider the process of work performed by an external force (W<0), then $T_{in}=T_{2}<T_{out}=T_{1}$ and:(3)$$\frac{W}{|Q_{2}|}\leq \frac{T_{2}-T_{1}}{T_{2}}<0.$$This correct condition applies to refrigeration processes in which, thanks to external work, it is possible to pump heat from a lower to a higher temperature. Such processes exhibit high coefficients of cooling efficiency $|Q_{2}|/|W|$ or the efficiency of the heat pump $|Q_{1}|/|W|\geq 1$. However, it is convenient to introduce an efficiency parameter $\tilde{\eta }\leq 1$ for the refrigeration processes, which corresponds to an engine process running in the opposite direction. Thus, such a parameter is determined analogously to (Equation 2), with the sense that the minuses are shortened $\tilde{\eta }=|W|/|Q_{1}|=\left(-W\right)/\left(-Q_{1}\right)=W/Q_{1}$. To find the condition for $\tilde{\eta }$, let us transform (Equation 3) by virtue of $W=|Q_{2}|-|Q_{1}|<0$:(4)$$1-\frac{|Q_{1}|}{|Q_{2}|}\leq 1-\frac{T_{1}}{T_{2}}\;\;\Rightarrow \;\;\frac{|Q_{2}|}{|Q_{1}|}\leq \frac{T_{2}}{T_{1}}\;\;\Rightarrow \;\;\tilde{\eta }\geq \eta_{C}.$$Thus, the efficiency parameter of $\tilde{\eta }$ cooling processes is not lower than the Carnot efficiency (unlike the engine processes which have the efficiency not greater than the Carnot efficiency). This result was obtained here on the basis of the application of the Carnot principle for negative work W<0. This result will be later deduced through independent considerations of a different kind.

Historically speaking, Carnot in 1824 [1] did not yet know the concept of absolute temperature and only defined the dependence of efficiency on a temperature difference [2]. Nevertheless, his considerations on the isothermal-adiabatic cycle implied the existence of maximum efficiency and minimal temperature. This minimum temperature along with the absolute temperature scale were formally introduced by Thomson (Lord Kelvin) [3]. Such absolute temperature scale is adopted in this work. This temperature is consistent with the zeroth law of thermodynamics, and its changes are consistent with the operation of a thermometer—so there is no need to give its elementary definition. The expression for the efficiency $\eta_{C}\left(T_{1},T_{2}\right)$ of the Carnot cycle was known as the Carnot function, which was not originally fully defined by Carnot. This function was intensely sought after by Clausius and Kelvin. Thus, the above-mentioned scholars, who had always referred to Carnot, also made an important contribution to the contemporary form of the C0 principle [4].

Carnot, as a pioneer of thermodynamics [3,5,6], did not know yet either the law of conservation of energy or the I law of thermodynamics [7], so he used the concept of the flow of the indestructible fluid of heat (caloric) [2]. Nevertheless, the Carnot principle can be considered in the context of the II law of thermodynamics [8,9,10], putting aside the fact that Carnot did not know the I law of thermodynamics. In the context of the [11] research of the finite-time Carnot cycle, it can be concluded that the idea of a Carnot engine, in a sense, is still not a closed concept in terms of engineering and experience.


**Clausius I principle (CI)**

*Heat naturally flows from a body at a higher temperature to a body at a lower temperature. Therefore, a direct (not forced by work) process of heat transfer from the body at a lower temperature to the body at a higher temperature is not possible.*
An absolute temperature scale is not needed to determine which reservoir has a higher temperature—even the Celsius scale is sufficient here. The Clausius I principle seems so trivial that one might suppose that it has no far-reaching, less trivial consequences. Nevertheless, it is allegedly proved, that the Clausius I principle is equivalent to the full II law of thermodynamics [12,13,14,15]. Criticism of this popular proof could only be found in a 2008 preprint by a scholar Bhattacharyya [16].

The CI principle is ascribed to Clausius’ publications from 1850, 1851 [17,18] and 1854, 1856 [19,20]. Contemporary work by Xue and Guo suggests that the above assignment is not strict [21]. These authors do not distinguish two Clausius principles (CI,CII), so it is not surprising that their reference to later Clausius textbooks [22,23] has to do with the second version of the Clausius principle (CII). However, in the chapter: *“III. Second Main Principle of the Mechanical Theory of Heat”* and in section: *“5. New Fundamental Principle concerning Heat”* of textbook [23], Clausius wrote: *“…and then enunciate the principle as follows: ‘A passage of heat from a colder to a hotter body cannot take place without compensation’. This proposition, laid down as a Fundamental Principle by the author…”* Such a declaration by Clausius from 1879 constitutes his principle in the first version (CI).



**Kelvin principle (K).**

*The processes of converting heat to work and work to heat do not run symmetrically. A full conversion of work to heat (internal energy) is possible. However, a full conversion of the heat to the work is not possible in a cyclical process.*
The principle called the Kelvin principle is formulated in various but similar ways. This relates not so much to the original formulation of Kelvin [24,25], but more to a later formulation of Planck (see [26]), called the Kelvin–Planck formulation (see [27]). The above version of the *K* principle reflects, as intended by the author, the essence of all these formulations—although this form of the principle comes from Planck or Ostwald ([26]) rather than initially from Kelvin. Nevertheless, traditional formulations identically equivalent to the *K* version of the principle are called the Kelvin principle, not the Kelvin–Planck or Ostwald principle [12,13,14,15,28,29,30,31]. In the publications [24,25,32], Kelvin expressed his principle as follows: *“It is impossible, by means of inanimate material agency, to derive mechanical effect from any portion of matter by cooling it below the temperature of the coldest of the surrounding objects”*. Such a formulation, however, seems weaker than the formulation of *K*, as it seems to refer to the performance of work at the expense of the cooler heat, and not more naturally at the expense of the heater heat. Therefore, it is not known which of the following elements Kelvin wanted to emphasize in this formulation: (i) a change in the temperature of the reservoir when taking heat converted into work, (ii) limitations in this conversion of heat to work, (iii) the impossibility of cooling down the coldest body by means of the work performed by this body? Thus, it can be said that Kelvin did not provide his own formulation of the II law of thermodynamics in a very strict and careful manner [2]. Therefore, it is not surprising that we use a version of the rule of the type *K* slightly different from the original formulation (see also [33]). Nevertheless, Kelvin’s formulation of the II law of thermodynamics is dated to 1851 in connection with his lectures published in the form of articles in two journals [24,25]. In addition, in the second journal, Kelvin separated a short article in which he repeated the formulation of his principle, and also gave a general interpretation of the II law of thermodynamics [32].


It can be concluded that the Kelvin principle in the *K* version (not original Kelvin version) is the same as the Ostwald principle, which says that there is no *perpetuum mobile* of type II ($\pi_{II}$) [26,28]. In other words, there is no heat engine with an efficiency of 100%:(5)$$\eta =\frac{W}{Q_{in}}=\frac{W}{Q_{1}}<1.$$It is assumed here that the heat source has a higher temperature than the heat receiver ($T_{in}=T_{1}>T_{out}=T_{2}$). Formally, however, it is possible to consider taking heat from a reservoir with a lower temperature (conventionally $Q_{2}<0$) and converting it into work W>0:(6)$$\eta_{2}=\frac{W}{|Q_{2}|}<1.$$And for the processes going in the opposite direction, the equivalent of this efficiency will have the form ${\tilde{\eta }}_{2}=|W|/|Q_{2}|$.

At first glance, the Kelvin principle seems to be very trivial and seems to be significantly weaker than the Carnot principle. Nevertheless, it is allegedly proved that the Kelvin principle in the form of *K* is equivalent to the full formulation of the II law of thermodynamics [12,13,14,15]. Criticism of this proof based on combining engine and heat pump processes could only be found in the aforementioned work by a scientist Bhattacharyya [16].



**Clausius II principle (CII)**

*There is a function of the state of the thermodynamic system called entropy, which the dependence of the value on time allows to determine the system’s following of the thermodynamic equilibrium. Namely, in thermally insulated systems, the only possible processes are those in which the total entropy of the system (two heat reservoirs) does not decrease:*

(7)
$$\Delta S=\Delta S_{1}+\Delta S_{2}\geq 0.$$


*For reversible processes, the increase in total entropy is zero, and for irreversible processes it is greater than zero. The change in entropy of a reservoir is here understood as the ratio of the heat introduced into it to its temperature.*
Thus, when using standard notations, Clausius II principle takes the following form for the processes considered here:(8)$$\Delta S=\frac{-Q_{1}}{T_{1}}+\frac{Q_{2}}{T_{2}}\geq 0.$$It is assumed that a specific device operating cyclically between heat reservoirs does not change its entropy by definition of cyclicity. Additionally, similarly for the C0 principle, the absolute temperature scale is assumed here.


The Clausius II principle is the most elegant formulation of the II law of thermodynamics. Some textbooks only introduce an entropy formulation of this principle (e.g., [34]). Only the definition of entropy presents some difficulty and challenge here. Clausius’s definition of entropy (integral of heat divided by temperature) is not always general enough in thermodynamics. However, we assume that it is sufficient for the process diagrams considered here. Another, usually overlooked, subtlety of the II law of thermodynamics (Clausius II principle and others) is that it actually speaks of states beyond the thermodynamic equilibrium ($T_{1}\neq T_{2}$) to which the system is approaching but which it does not necessarily achieve (e.g., on diagrams with unlimited heat reservoirs—thermostats).

The year 1865 of the publication of work [35] is regarded to be the year when principle CII was created, even though in the years 1854 and 1856 in Clausisus works [19,20] there were expressions of the type N=∑Q/T or N=∫dQ/T with the correct analysis of signs (N>0 for irreversible processes, N<0 for impossible and N=0 for reversible). The use of inequalities in physics would be something completely new, so in the 1850s in Clausius works they do not appear explicitly. Inequalities in the mathematical formula can be found in Clausius, for example, in the textbook [22] from 1867. In this textbook, Clausius also changes the notation of *N* to *S* by calling this quantity a transformation value (process measure). Then, translating *transformation* into Greek (τρoπη), Clausius introduces the term *entropy*. Of course, Clausius distinguished the entropy change in a process from its absolute value. The concept of entropy entered the canon of physics, and the principle of its growth (more precisely, its not decreasing) constitutes, for example, the term *entropy production* [36]. This form of the principle is valid in elementary systems of statistical mechanics [37].

There are also other attempts to formulate the II law of thermodynamics. The formulation by Caratheodory is sometimes given: *“In the surroundings of each thermodynamic state, there are states that cannot be achieved by adiabatic processes”*. However, such a formulation is expressed in a completely different language than that considered in this work. Moreover, this statement sounds as tautological as the statement: *there are places in the mountains that are inaccessible to hiking trails*. At the same time, Caratheodory’s statement is given very often without strict proof of equivalence with other formulations (see [29,31]). From a contemporary critique of Caratheodory’s formulation in preprint [38] we also learn that this formulation was greatly criticized by Planck.

The second law of thermodynamics is also formulated in the language of statistical physics. The outstanding physicist Ludwig Boltzmann was the first to undertake this task. Unfortunately, this article is limited to phenomenological thermodynamics only.

In the context of thermodynamic considerations, the abstract concept of *perpetuum mobile* often appears. It is, therefore, worthwhile to clarify and differentiate this concept as follows.


***Perpetuum mobile*** **($\pi_{0},\pi_{I},\pi_{\mathrm{II}},\pi_{\mathrm{III}}$)**
*A perpetuum mobile is a hypothetical machine, the movement and efficiency of which would contradict certain recognized laws of physics. The most faithful meaning of the Latin name (“forever moving”) perpetuum mobile type 0 ($\pi_{0}$) would not be stopped in spite of the existence of frictional forces and resistance to motion. A perpetuum mobile type I ($\pi_{I}$) would do work from nothing (or almost nothing—efficiency greater than 100%) against the principle of conservation of energy and against the I law of thermodynamics. A perpetuum mobile type II ($\pi_{II}$) would be 100% efficient and would be against the Kelvin principle. On the other hand, the perpetuum mobile type III ($\pi_{III}$) would have efficiency lower than 100%, but higher than the maximum efficiency predicted by the Carnot principle.*
Most often it only stands out *perpetuum mobile* $\pi_{I}$ and $\pi_{II}$. However, *perpetuum mobile*
$\pi_{III}$ is crucial for this article. Sometimes the term *perpetuum mobile* of type III is understood as $\pi_{0}$. However, the proposed terminology seems clear and consistent with the meaning of the Latin words *perpetuum mobile* for $\pi_{0}$ and consistent with the classification of various ways of doing work by $\pi_{I},\pi_{II},\pi_{III}$—while machine $\pi_{0}$ does not do the work. A similar notation to *perpetuum motion* will also be applied to the $\pi^{↑}$ process where heat spontaneously flows uphill (towards higher temperature). Such a process, despite its similarities, does not have to be automatically equivalent to $\pi_{II}$.


Laws of physics, such as the thermodynamic principles under consideration, are subject to some kind of experimental verification. Unfortunately, from a formal point of view, these principles are not subject to truth proof like mathematical theorems. Nevertheless, the two principles may result from each other. Unfortunately, the implication of principles is something more complex than the material implication of Table 1, that depends only on logical values.

Two principles may be subject to a relationship of formal implication. The formal implication related to a specific theoretical system implemented by certain models becomes a semantic implication. A provable formal implication is called syntactic implication. Kurt Gödel proved the theorem that a semantic implication is a syntactic implication. In other words, a model-based system is needed to prove formal implication. In the case of thermodynamic principles, models are sets of processes (diagrams) that follow a given principle. Thus, the formal implication of the two principles can be written as follows:(9)$$Z_{1}|\;\to Z_{2}\Leftrightarrow \forall d\in \Omega :\;\;d\in \left\{Z_{1}\right\}\Rightarrow d\in \left\{Z_{2}\right\},$$
where {Z} stands for the model, i.e., the set of processes consistent with the *Z* principle. If there are several (or many) models, the formal implication should apply for all $Z_{1}$ models, and there should be at least one $Z_{2}$ model containing the same *d* process. The symbol of formal (syntactic) implication ∣→ used here was created from the combination of the symbol known from the formal logic ⊢ and the usual symbol of the implication arrow → (see [39]). Strictly speaking, this symbol in LaTeX transcription is a combination of the sign ∣ with the sign → using a negative space (“∖med ∖!∖rightarrow”). In formal logic, the symbol ⊧ is also used for the semantic formal implication. In model theory, this symbol is synonymous, but it is used to constitute a sentence, thesis, or a principle within some system [40]. For example, the fact that in the system of theoretical thermodynamics T the Kelvin law is true we write as T⊧K or more precisely T⊧(CII∣→K), where we assume the truth of the CII principle.

We can see that the system of models (sets of processes) allows us to reduce the formal implications to the material implications. The significant difference, however, is that the material implication must hold true for all the processes (diagrams) of the system to which the formal implication relates. This also applies to transposition law, which is used in the scheme of proof by contradiction:(10)$$Z_{1}|\;\to Z_{2}\Leftrightarrow \forall d\in \Omega :\;\;d\notin \left\{Z_{2}\right\}\Rightarrow d\notin \left\{Z_{1}\right\}.$$

Thus, in order to prove the implication of the principles, it is not enough to point to the $d^{\prime }$ process, which contradict the $Z_{2}$ principle, and to show that it implies a contradiction with the $Z_{1}$ principle (compare Figure 2a with Figure 2b). It is also worth noting that proof by contradiction is not the most valued method of proof. Constructive proof that follows the original direction of implication is the most valued. There is even an orthodox version of logic (intuitionistic logic), that rejects the law of the excluded middle—including proof by contradiction. There have even been attempts to integrate intuitionism into the mathematical proving system. The Brouwer’s program is the best known in this respect.

So, in order to prove the $Z_{1}|\;\to Z_{2}$ implication of the principles, it is necessary to show that the sets of processes conforming to these principles (their models) are subsets $\left\{Z_{1}\right\}\subset \left\{Z_{2}\right\}$. The set of the resulting (often weaker) principle should, therefore, be a set greater than or equal to the set of the previous (often stronger) principle. It is a little easier to rebut the implications of two principles. For this purpose, it is enough to indicate the $d_{1}$ process, which belongs to the $\left\{Z_{1}\right\}$ set, and does not belong to the $\left\{Z_{2}\right\}$ set (see Figure 2b).

The principles are formally equivalent when the formal implication acts both ways:(11)$$Z_{1}↔Z_{2}\Leftrightarrow \left(Z_{1}|\;\to Z_{2}\right)\wedge \left(Z_{2}|\;\to Z_{1}\right)\Leftrightarrow \forall d\in \Omega :\;\;\left(d\in \left\{Z_{1}\right\}\Leftrightarrow d\in \left\{Z_{2}\right\}\right).$$ Usually the equivalence of the three principles is proved by a looped chain of implications $Z_{1}|\;\to Z_{2}|\;\to Z_{3}|\;\to Z_{1}$ (or just as well in the opposite direction). However, this way of proving is prone to error—it is enough to undermine the weakest link in the chain loop for the equivalence proof to be invalid. A more comprehensive approach is to analyze the entire matrix of implications (Table 2).

We can see that of the 12 implications, 3 do not have a clear *status quo*. The point is that Carnot’s principle is not given as an equivalent formulation of the II law of thermodynamics, but as a consequence of this law. Moreover, we see that as many as 6 other implications are being questioned. The point is that the author supposes that the principles of Clausius I and Kelvin are not equivalent to each other and are not equivalent to the II law of thermodynamics. The main goal of this paper is to formally prove these hypotheses.

## 2. Materials and Methods

The work is theoretical in nature, based on strict proof of the formal implications of the relevant principles and the rebuttals of the remaining formal implications, which are not satisfied. The considered formal implications relate to the defined simplified conceptual system of thermodynamics. This system consists of a set of Ω of all real or virtual processes (diagrams), along with some elementary operating rules for these processes.

An exemplary engine diagram is shown in Figure 3. For convenience, such a diagram will be marked with the arrows ↓→↓, informing respectively that: heat flows from the heater, work is performed by the device (engine), heat flows to the cooler. The arrows of such a diagram determine the signs of heat and work in the associated record of the $\left(Q_{1},W,Q_{2}\right)$ process, which in this case of the engine process means the positivity of all parameters: $Q_{1}>0,W>0,Q_{2}>0$. If an arrow is pointing in the opposite direction, it means that the part of the process is going the opposite way and the corresponding heat or work symbol is negative (Figure 4). However, if a given part of the process does not occur, it is marked with 0, e.g., diagram ↓0↓ represents a process in which heat flows from the heater to the cooler without any work being done. It is assumed that the symbols of the diagrams represent (apart from the null 000 process) a continuum of processes with different values of heat and work following the arrows (and the I law of thermodynamics). In addition, diagrams specifying the efficiency of the engine $(↓\to {↓)}_{\eta }$ or the efficiency of the heat pump $(↑\leftarrow {↑)}_{\tilde{\eta }}$ will also be considered.

As mentioned before, *a priori* is assumed that the diagrams satisfy the law of conservation of energy, or the I law of thermodynamics: $Q_{1}=W+Q_{2}$. Additionally, in the diagram set, a rule for adding diagrams is introduced, which physically means linking the processes. Adding diagrams is used, among other things, to define the concept of a convex of a set of diagrams:**The definition of a convexity (for a set of diagrams—for a model)***A set of diagrams will be called convex if the sum of any two diagrams of this set belongs to this set. In other words, the convexity of the set requires that the add diagrams operation be internal to that set.*Thus, the definition of the convexity of the set is here simplified to the operation of adding elements without having to consider multiplying diagrams by non-negative real numbers. However, the ability to scale diagrams means that the above definition is equivalent to the standard geometric definition of a convex set. The definition of a convex set helps to formulate the following important condition:**Completeness condition (for model of principle)***It is assumed that the set of diagrams (processes), consistent with a given principle and called the model of this principle, must be the largest convex set possible. It is allowed to have many alternative sets (models) that meet the above condition, in the sense that a given set (model) cannot be enlarged without breaking compliance with the principle or without breaking the convexity requirement.*The completeness condition allows for the consideration of richer models (sets of diagrams) for a given principle. Thanks to this condition, diagrams (processes) whose compliance with the principle is not obvious can be attached to the model. An attached diagram (process) can be created as a result of adding two trivial diagrams or diagrams already belonging to the model. In addition, a diagram (process) can be attached to the model, the addition of which to existing diagrams results in a trivial diagram or a diagram belonging to the model. This procedure can fork the model into two models or the entire model family. Regardless of the number of models in the family, each model should be considered an alternative independent model. If there is only one model, it will be called a homogeneous model here, and in the literature it is sometimes called a stable model.

Given the above theoretical structure, we can study the formal implications (or lack thereof) between the four elementary principles of thermodynamics (C0,CI,K,CII). There are as many as 12 formal implications from Table 2 to be proved or disproved. Proof of the $Z_{1}|\;\to Z_{2}$ implication will be performed double, once by checking the content of the sets of diagrams (models) of both principles $\left\{Z_{1}\right\}\subset \left\{Z_{2}\right\}$, two by formal constructive proof (proof by contradiction will not be used). Similarly, disproving the implications will consist, firstly, in checking that the models do not contain themselves $\left\{Z_{1}\right\}⊄\left\{Z_{2}\right\}$, and secondly, in pointing to a counterexample in the form of a diagram $d_{1}$ consistent with principle $Z_{1}$, but inconsistent with principle $Z_{2}$ (see Figure 2b). The negation of the formal implication will be denoted by the standard negation operator $\neg (Z_{1}|\;\to Z_{2})$ or by the crossed arrow $Z_{1}|\;↛Z_{2}$.

The proof and rebuttals will be presented in the Discussion section and summarized in the Results section. A synthesis will be given in the Conclusions.

## 3. Discussion: Principles Structure, Proof and Rebuttals

Before starting the analysis of formal implications between thermodynamic principles, it is worth writing out sets of diagrams (models) for individual principles. Some principles have a complex structure of models that take into account different possible scenarios for realizing a given principle.

### 3.1. Structure of the Carnot Principle Model

The main processes mentioned in the Carnot principle are ↓→↓ engine processes with efficiency not higher than $\eta_{C}$. This set can be extended to include limit cases in the form of processes: zero efficiency ↓0↓ and null process 000.

The remainder of the model will be found by eliminating processes, that are inconsistent with the condition of internality of adding processes. Consider the $(↓\to {↓)}_{\eta_{C}}$ engine process with a maximum Carnot efficiency of $\eta_{C}$. Let us describe the parameters of this non-zero process with three values $\left(Q_{1},W,Q_{2}\right)$. Less efficient processes have the form $\left(Q_{1}+q,W,Q_{2}+q\right)$, where q>0. Note that such processes are the sum of the Carnot process with the maximum efficiency $(↓\to {↓)}_{\eta_{C}}$ and the process with zero efficiency ↓0↓(q,0,q). On the other hand, adding the ↑0↑(−q,0,−q) process to the Carnot process would lead to obtaining a virtual (impossible) process with efficiency greater than Carnot efficiency. Thus, processes of type ↑0↑ should be excluded from the Carnot principle model. Similarly, we will exclude type $↑\leftarrow ↑(-Q_{1}-q,-W,-Q_{2}-q)$ refrigeration processes, which are the inverse of engine processes with less than maximum efficiency. Well, by adding the Carnot process to this process, we get a process outside the model (q>0):(12)$$\left(-Q_{1}-q,-W,-Q_{2}-q\right)+\left(Q_{1},W,Q_{2}\right)=\left(-q,0,-q\right)\notin \left\{C0\right\}.$$

In addition to the excluded refrigeration processes, there are form $↑\leftarrow ↑(-Q_{1}+p,-W,-Q_{2}+p)$ refrigeration processes which are not excluded for p≥0. It is most convenient to assign the efficiency parameter $\tilde{\eta }$ to such refrigeration processes, which refers to the inverse engine process. Although the inverse engine process is virtual ($\tilde{\eta }\geq \eta_{C}$), the refrigeration process $(↑\leftarrow {↑)}_{\tilde{\eta }}$ exists and belongs to the Carnot principle model.

A borderline case of the ineffective refrigeration process of ↑←↑ for $\tilde{\eta }\to 1$ is the process of converting work into heat of the heater ↑←0. The superposition of the ↑←0 process with an appropriately selected ↓0↓ process gives the 0←↓ process. And combining the last two processes results in a new type of process ↓←↓. However, as we add a small heat flow in the form ↓0↓ to the ↑←0 process, we get ↑←↓.

In summary, the Carnot principle model consists of the following processes:(13)$${\left\{C0\right\}=\{000,\;↓0↓,\;0\leftarrow ↓,\;↑\leftarrow 0,\;↓\leftarrow ↓,\;↑\leftarrow ↓,\;\;(↓\to ↓)}_{0<\eta \leq \eta_{C}},(↑\leftarrow {↑)}_{\eta_{C}\leq \tilde{\eta }<1}\}.$$ We can see that the uniform (stable) model of Carnot principle consists of diagrams of 8 types. There are 3×3×3=27 possible all diagram types (make sense or not ). However, some diagrams contradict the principle of conservation of energy: with two zeros 3×2=6, with one zero 3×2=6, without zeros 2—which adds up to 6+6+2=14 for diagrams outside areas of consideration. So in the domain of consideration there are 27−14=13 of diagram types. This means that the Carnot principle model rejects five types of diagrams.

Therefore, it is worth listing the complement to the Carnot model (a set of impossible processes):(14)$$\bar{\left\{C0\right\}}{=\{↑0↑,\;0\to ↑,\;↓\to 0,\;↑\to ↑,\;↓\to ↑,\;\;(↓\to ↓)}_{\eta_{C}<\eta <1},(↑\leftarrow {↑)}_{0<\tilde{\eta }<\eta_{C}}\}.$$ Indeed, five new types of diagrams were created simply by changing the direction of the corresponding arrows. The other two diagrams do not differ in type but have different ranges of efficiency values. Thus, combinatorics is here in line with the resulting model of Carnot’s principle.

### 3.2. Structure of Models for Clausius I Principle

The Clausius principle I has a whole family of models, that can be parameterized with the maximum efficiency $\eta_{m}\in \left[0,1\right]$:(15)$$\left\{CI\right\}={\left\{CI\right\}}_{\eta_{m}}={\left\{CI\right\}}_{0},\;{\left\{CI\right\}}_{0<\eta_{m}<1},\;{\left\{CI\right\}}_{1}.$$ We have a separate principle model for each value of maximum efficiency. The principle allows for both a zero efficiency model containing only refrigeration processes without engine processes, and also allows for an efficiency model of 1 (*perpetuum mobile* $\pi_{II}$) without refrigeration processes:(16)$${\left\{CI\right\}}_{0}=\left\{000,\;↓0↓,\;0\leftarrow ↓,\;↑\leftarrow 0,\;↓\leftarrow ↓,\;↑\leftarrow ↓,\;no\;engines,\;↑\leftarrow ↑\right\},$$
(17)$${\left\{CI\right\}}_{0<\eta_{m}<1}{=\{000,\;↓0↓,\;0\leftarrow ↓,\;↑\leftarrow 0,\;↓\leftarrow ↓,\;↑\leftarrow ↓,\;(↓\to ↓)}_{0<\eta \leq \eta_{m}},(↑\leftarrow {↑)}_{\eta_{m}\leq \tilde{\eta }<1}\},$$
(18)$${\left\{CI\right\}}_{1}=\left\{000,\;↓0↓,\;0\leftarrow ↓,\;↑\leftarrow 0,\;↓\leftarrow ↓,\;↑\leftarrow ↓,\;no\;refrigerators,\;↓\to 0,\;↓\to ↓\right\}.$$ The first six trivial processes are common to all models, but the engine and refrigeration processes already differ in the sense that in extreme cases one of them does not occur.

### 3.3. Structure of Models for Kelvin Principle

The structure of the Kelvin principle models is the most complex of all the principles considered in this paper. It turns out that this principle allows scenarios of models with both “top-down” ↓ and “bottom-up” ↑ heat flows. The zero-efficiency scenario model allows for both directions simultaneously. Thus, the structure of the Kelvin models can be represented as follows:(19)$$\left\{K\right\}={\left\{K\right\}}_{0}^{↕},\;{\left\{K\right\}}_{0<\eta_{m}<1}^{↓},\;{\left\{K\right\}}_{1^{-}}^{↓},\;{\left\{K\right\}}_{0<\eta_{m}<1}^{↑},\;{\left\{K\right\}}_{1^{-}}^{↑}.$$

Models containing processes of arbitrarily high efficiency but less than one ($1^{-}$), which should not be confused with models with a specified maximum efficiency of $\eta_{m}$, have been distinguished here. The zero-efficiency model is as follows:(20)$${\left\{K\right\}}_{0}^{↕}=\left\{000,\;↓0↓,\;0\leftarrow ↓,\;↑\leftarrow 0,\;↓\leftarrow ↓,\;↑\leftarrow ↓,\;no\;engines,\;↑\leftarrow ↑,\;↑0↑\right\}.$$ Models with a natural direction of heat flow (“downwards”) and with an intermediate maximum efficiency coincide with the analogous models for the Clausius I principle:(21)$${\left\{K\right\}}_{0<\eta_{m}<1}^{↓}{=\{000,\;↓0↓,\;0\leftarrow ↓,\;↑\leftarrow 0,\;↓\leftarrow ↓,\;↑\leftarrow ↓,\;(↓\to ↓)}_{0<\eta \leq \eta_{m}},(↑\leftarrow {↑)}_{\eta_{m}\leq \tilde{\eta }<1}\}.$$ On the other hand, the model containing the efficiencies arbitrarily close to one, differs from the analogous model of the CI principle only by the lack of processes with the efficiency of one:(22)$${\left\{K\right\}}_{1^{-}}^{↓}=\left\{000,\;↓0↓,\;0\leftarrow ↓,\;↑\leftarrow 0,\;↓\leftarrow ↓,\;↑\leftarrow ↓,\;no\;refrigerators,\;↓\to ↓\right\}.$$ In models with the opposite direction of heat flow, it is enough to reverse the directions of the corresponding arrows:(23)$${\left\{K\right\}}_{0<\eta_{m}<1}^{↑}{=\{000,\;↑0↑,\;0\leftarrow ↓,\;↑\leftarrow 0,\;↑\leftarrow ↑,\;↓\leftarrow ↑,\;(↑\to ↑)}_{0<\eta_{2}\leq \eta_{m}},(↓\leftarrow {↓)}_{\eta_{m}\leq {\tilde{\eta }}_{2}<1}\},$$
(24)$${\left\{K\right\}}_{1^{-}}^{↑}=\left\{000,\;↑0↑,\;0\leftarrow ↓,\;↑\leftarrow 0,\;↑\leftarrow ↑,\;↓\leftarrow ↑,\;no\;"refrigerators",\;↑\to ↑\right\}.$$ The term “*refrigerators*” is given here in quotation marks, because the heat in such a device would be pumped from a higher temperature to a lower temperature, in these models. It is worth noting that these models do not have the usual engines (*no engines*), i.e., processes in which the work is done at the expense of the heat of the heater, not the cooler. The lack of typical engines ensures the consistency of these models. Such a class of models obviously contradicts the tendency towards thermodynamic equilibrium, but the Kelvin principle does not seem to determine this condition.

Above all, however, both classes of heat flow direction, similarly to the Clausius I principle, have the efficiency values $\eta_{m}$, which may exceed the Carnot efficiency $\eta_{C}$ ($\eta_{m}>\eta_{C}$). The value of $\eta_{m}<1$ is not related in any way to the value of $\eta_{C}$. Hence, we have whole classes (sets families) of Kelvin (and Clausius I) models. Models for which $\eta_{m}>\eta_{C}$ contain more engine diagrams than the uniform (stable) and common model for the Carnot principle and Claussius II principle.

### 3.4. Structure of the Clausius II Principle Model

The principle CII about non-decreasing entropy of physical processes is consistent with adding processes and the condition of completeness of the model. In other words, the compliance of processes with the CII principle automatically guarantees the internality of adding processes. As a result, the model of the Clausius II principle is a homogeneous single set (stable model):(25)$${\left\{CII\right\}=\{000,\;↓0↓,\;0\leftarrow ↓,\;↑\leftarrow 0,\;↓\leftarrow ↓,\;↑\leftarrow ↓,\;\;(↓\to ↓)}_{0<\eta \leq \eta_{C}},(↑\leftarrow {↑)}_{\eta_{C}\leq \tilde{\eta }\leq 1}\}.$$ The above model closely follows the Carnot principle model:(26){CII}={C0},
but it is narrower than the families of models for Clausius I and Kelvin principles. Strictly speaking, the {CII} model coincides with only one model ${\left\{CI\right\}}_{\eta_{m}=\eta_{C}}$ of Clausius I principle and only one model ${\left\{K\right\}}_{\eta_{m}=\eta_{C}}^{↓}$ of Kelvin principle. Therefore, Clausius II principle and the equivalent Carnot principle are stronger principles than those of Clausius I and Kelvin.

The structure of the (Equation 13), (Equation 15), (Equation 19), (Equation 25) models already specifies the table of all implications or their absence. However, these implications will be subject to a more detailed proof analysis.

First, the implications of the remaining principles from the Carnot principle will be analyzed.

### 3.5. Theorem C0∣→CI

**Proof.** Based on (Equation 13) and (Equation 15), (Equation 17) we can see that $\left\{C0\right\}\subset {\left\{CI\right\}}_{\eta_{m}}$ for $\eta_{m}=\eta_{C}$. Consider a Carnot heat engine denoted by the diagram $(↓\to {↓)}_{\eta_{C}}$, in which heat input, work done and output heat can be written as $\left(Q_{1},W,Q_{2}\right)$. Similarly, we will write the process of heat flow (q,0,q), which is a flow down (towards a lower temperature) ↓0↓ for q>0 or a flow up (towards a higher temperature) ↑0↑ for q<0. Now let us combine these two processes:
(27)$$\left(Q_{1},W,Q_{2}\right)+\left(q,0,q\right)=\left(Q_{1}+q,W,Q_{2}+q\right).$$ Let us only consider the resultant processes which follow the Carnot principle:
(28)$$\eta =\frac{W}{Q_{1}+q}\leq \frac{W}{Q_{1}}=\eta_{C}\;\;\;\Rightarrow \;\;\;q\geq 0.$$ Therefore, on the basis of the completeness condition, we can conclude that the Carnot principle is compatible with the processes of heat flow downwards ↓0↓(q>0), and the processes of heat flow upwards ↑0↑(q<0) do not occur. Thus, we obtained the thesis of Clausius I principle. □

### 3.6. Theorem C0∣→K

**Proof.** Based on (Equation 13), (Equation 19) and (Equation 21) we can see that $\left\{C0\right\}\subset {\left\{K\right\}}_{\eta_{m}}^{↓}$ for $\eta_{m}=\eta_{C}$. Since essentially $\left\{K\right\}={\left\{K\right\}}_{0}^{↕},{\left\{K\right\}}^{↓},{\left\{K\right\}}^{↑}$, it can be assumed that {C0}⊂{K}. Thus, the set of Carnot principle processes follows the Kelvin principle. These are, of course, processes with the natural direction of heat flow downwards (as in principle of Clausius I).Consider a process involving an ↓→? heat engine in which we do not know if the cooler absorbs heat ($Q_{2}=?$). Let us describe this process with three parameters $\left(Q_{1},W,Q_{2}\right)$, where $Q_{1}>0,W>0,Q_{2}=Q_{1}-W$. The Carnot principle implies the condition for $Q_{2}$:
(29)$$\eta =1-\frac{Q_{2}}{Q_{1}}\leq \eta_{C}<1\;\;\;\Rightarrow \;\;\;Q_{2}\geq \left(1-\eta_{C}\right)Q_{1}>0.$$ This means that heat must be transferred to the cooler, so it is a ↓→↓ process with less than 1 efficiency, which is in line with the Kelvin principle. □

### 3.7. Theorem C0∣→CII

**Proof.** Based on (Equation 13) and (Equation 25) we can see that {C0}={CII}, so in particular also {C0}⊂{CII}. Thus, on the basis of the models (sets of processes), there is implication (or even equivalence).Consider the Carnot engine process $↓\to ↓(Q_{1},W,Q_{2})$ and the natural heat flow downward process ↓0↓(q,0,q). On the basis of (Equation 28) within the Carnot principle we know that the process inverse to the second process cannot take place (so q≥0). The completeness condition implies the existence of a summary process $\left(Q_{1}+q,W,Q_{2}+q\right)$. Let us calculate the entropy change of the heat reservoirs in this process:
(30)$$\Delta S=\frac{-Q_{1}-q}{T_{1}}+\frac{Q_{2}+q}{T_{2}}=0+\frac{T_{1}-T_{2}}{T_{1}T_{2}}q\geq 0.$$ which is in line with the Claussius II principle.Now consider the Carnot refrigeration process $↑\leftarrow ↑(-Q_{1},-W,-Q_{2})$. By adding the ↓0↓(q,0,q) process to it, we get the allowed $\left(-Q_{1}+q,-W,-Q_{2}+q\right)$ process. The change in the entropy of the heater and cooler in this refrigeration process is:
(31)$$\Delta S=\frac{Q_{1}-q}{T_{1}}+\frac{-Q_{2}+q}{T_{2}}=0+\frac{T_{1}-T_{2}}{T_{1}T_{2}}q\geq 0.$$ which, again, is in line with the Clausius II principle. The change of entropy would be negative for the $\left(-Q_{1}-q,-W,-Q_{2}-q\right)$ refrigeration process—however, such a process is not compatible with the Carnot principle, as the convexity condition allows for addition, not subtraction, of process diagrams.Equality of process sets of the C0 and CII principles and the fulfillment of the CII principle for processes permitted by the C0 principle ends the proof. □

Now we turn to the implications of Clausius I principle. The first implication, however, does not hold true (as do the other two).

### 3.8. Theorem ¬(CI∣→C0)

**Proof.** Based on (Equation 13) and (Equation 15) we can see that the model family for the CI principle is larger than the uniform set of processes (stable model) for the C0 principle. These are additional models that contain engine processes with an efficiency greater than that of the Carnot cycle ($\eta_{C}<\eta_{m}<1$). In addition, there is even a ${\left\{CI\right\}}_{1}$ model containing *perpetuum mobile* type II, which is contrary to the Carnot principle.First of all, it should be shown that processes with the maximum efficiency in the range $\eta_{C}<\eta_{m}<1$ are allowed by the CI principle. Although this principle distinguishes between higher and lower temperatures and the direction of heat flow, it does not impose quantitative restrictions on efficiency (even a model with $\eta_{m}=1$ is possible).Consider the Carnot engine cycle process $↓\to ↓(Q_{1},W,Q_{2})$. We assume that this process complies with the Clausius I principle. Let us consider whether there is a possible model of the CI principle, in which there is a process with even greater efficiency, i.e., process $↓\to ↓(Q_{1}-q,W,Q_{2}-q)$, where q>0. The inclusion of such a process will not lead to a contradiction, if only some refrigeration processes exist in the model. For example, $↑\leftarrow ↑(-Q_{1}+q+p,-W,-Q_{2}+q+p)$ refrigeration processes will be acceptable for p≥0. Their addition to the Carnot cycle process and their addition to the more efficient process under consideration leads to downward heat flow processes ↓0↓ or to a null process 000, which are in accordance with the CI principle. In the considered model of the CI principle, however, the refrigeration process type $↑\leftarrow ↑(-Q_{1},-W,-Q_{2})$ cannot occur because the higher efficiency engine added to the process would generate a heat flow upwards ↑0↑ contrary to the CI principle. However, the absence of $\left(-Q_{1},-W,-Q_{2}\right)$ in the model does not make it contradictory. Thus, the considered model of the CI rule is acceptable. At the same time, this model of the CI principle is not compatible with the model of the C0 principle, as it contains engine processes with efficiency greater than the Carnot cycle. □

### 3.9. Theorem ¬(CI∣→K)

**Proof.** Based on (Equation 15), (Equation 18) and (Equation 19), (Equation 22) we can see that {CI}⊄{K}. The absence of inclusions here is quite subtle and is due to the lack of inclusions of special models ${\left\{CI\right\}}_{1}⊄{\left\{K\right\}}_{1^{-}}^{↓}$. This is due to the existence of a *perpetuum mobile* process of the II type ${↓\to 0\in \left\{CI\right\}}_{1}$, which is not allowed by the Kelvin principle ${↓\to 0\notin \left\{K\right\}}_{1^{-}}^{↓}$.Thus, the rebuttal of the implication in this case is based on the existence of one counterexample: the ↓→0 process in one model of Clausius I principle. It is about the ${\left\{CI\right\}}_{1}$ model the processes of which are listed in (Equation 18). The ↓→0 process in this model does not lead to a contradiction because there are no refrigeration processes in this model that could lead to a contradiction. Refrigeration processes may exist in other models, however, where there is no process ↓→0. The existence of the ${\left\{CI\right\}}_{1}$ model removes the formal implication between the CI principle and the *K* principle.It is also necessary to point out the error in the alleged contemporary proof by the contradiction of the considered implication. Consider two processes that contradict Kelvin principle: ↓→0(W,W,0) and 0→↑(0,W,−W), whose efficiency is 100%. Let us assume a non-zero refrigeration process $↑\leftarrow ↑(-Q_{1},-W,-Q_{2})$. Combining such a refrigeration process with the above 100% efficiency processes leads to upward heat flow processes, respectively: $↑0↑(-Q_{2},0,-Q_{2})$ and $↑0↑(-Q_{1},0,-Q_{1})$. Thus, it might seem that breaking Kelvin principle entails breaking Clausius I principle (alleged proof by contradiction). However, it is not so. First, the consideration of the ↑←↑ refrigeration process is consistent with, and not in contradiction to, the Kelvin Principle—upon which proof by contradiction should be based. Not surprisingly, considering contradicting the Kelvin principle and adopting the Kelvin principle leads to a contradiction. Second, there is nothing to prove that the combination of a virtual process (outside the model) with a process within the model that leads to a process outside the model (one or the other principle). The condition for the internality of adding processes in the completeness condition applies to the situation when both the processes being added belong to the model. Third, the alleged proof by contradiction relies on a material implication for two cases, while it should be based on a formal implication based on all possible processes. □

### 3.10. Theorem ¬(CI∣→CII)

**Proof.** Based on (Equation 15) and (Equation 25) we can see that {CI}⊄{CII}. From the whole family of $\eta_{m}\in \left[0,1\right]$ models for the CI principle, only the $\eta_{m}=\eta_{C}$ model coincides with the CII principle model. Even the zero-efficiency $\eta_{m}=0$ model is not included in the {CII} model because it contains too many refrigeration processes. Thus, there is not formal implication from the CI principle to the CII principle.Consider two counterexamples of implications in the form of processes, that are allowed in the CI principle models but not in the CII principle model. The first counterexample would be *perpetuum mobile* type II ↓→0, which was already given in the previous section in the context of the CI rule. The second counterexample should be the efficient refrigeration process $\left(-Q_{1}-q,-W,-Q_{2}-q\right)$ related to the Carnot cycle $\left(Q_{1},W,Q_{2}\right)$. It is easy to verify that for q>0 the refrigeration process would increase the entropy of the heat reservoirs (ΔS>0), so it does not follow the CII rule. However, this refrigeration process will follow the CI principle model, which does not include the Carnot cycle, but the less efficient cycle $\left(Q_{1}+q,W,Q_{2}+q\right)$. Then adding these processes does not lead to contradiction with the model for the CI principle. The indicated counterexamples disproves the implication CI∣→CII.Errors in the alleged proof of the disproved implication will now be shown. Consider the virtual engine process $↓\to ↓(Q_{1}-q,W,Q_{2}-q)$ with efficiency greater than that of a Carnot engine $↓\to ↓(Q_{1},W,Q_{2})$ for q>0. Such a process obviously increases the entropy of the system and is against the CII principle. Additionally, consider the Carnot refrigeration cycle $↑\leftarrow ↑(-Q_{1},-W,-Q_{2})$. The superposition of the virtual engine cycle with the Carnot refrigeration cycle gives:
(32)$$\left(Q_{1}-q,W,Q_{2}-q\right)+\left(-Q_{1},-W,-Q_{2}-q\right)=\left(-q,0,-q\right),$$
which is contrary to the CI principle of the upward heat flow process ↑0↑. One might think that this is proof by contradiction for the implication under consideration. However, nothing could be more wrong. Firstly, the simultaneous use of the process contrary to the CII principle and the second process consistent with the principles of CI and CII raises doubts as to whether the CII principle was actually negated (in the proof by contradiction). Secondly, the proof should apply to all processes, not just one process (material implication vs formal implication).Thus, based on the lack of inclusion of models, two counterexamples, and an error indication in the supposed typical proof, the implication CI∣→CII is rebutted. □

It will now be shown that the other principles do not formally follow from the Kelvin principle.

### 3.11. Theorem ¬(K∣→C0)

**Proof.** Based on (Equation 13) and (Equation 19) we can see that {K}⊄{C0}. The *K* principle has rich family of models, so they cannot be contained in a single model for the C0 principle. At the level of the model structure, the lack of the considered formal implication is quite clear.However, let us give two counterexamples of processes that conform to the *K* rule, but not the C0 rule. Let it be the $↓\to ↓(Q_{1}-q,W,Q_{2}-q)$ engine process with any high efficiency less than 100 % ($q<Q_{2},q\approx Q_{2}$) and the ↑0↑ process heat flow upward (towards higher temperature). These processes are found in the models of *K* principle and do not appear in the model of C0 principle. Contrary to appearances, the inverse Carnot cycle $\left(-Q_{1},-W,-Q_{2}\right)$ cannot be added to the engine process in order to obtain a contradictory process ↑0↑ for these models. This cannot be done because the ultra high efficiency model does not have cooling cycles for $\tilde{\eta }=\eta_{C}$. However, when it comes to the ↑0↑ process, it is possible in separate models that implement the scenario of spontaneous heat flow upward (towards a higher temperature). The Kelvin Principle does not speak of heat reservoirs temperatures, so it should come as no surprise that it determines neither the heat direction nor the Carnot efficiency. □

### 3.12. Theorem ¬(K∣→CI)

**Proof.** Based on (Equation 19) and (Equation 15) we can see that {K}⊄{CI}. Both principles have large families of models, but the Kelvin family of models is larger with models with opposite heat flow directions.Counterexamples for the considered implication may be the process of heat flow upward ↑0↑ or a process of finite efficiency ↑→↑ of heat conversion from the lower temperature reservoir to work. Both processes are simultaneously included in the two models ${\left\{K\right\}}_{0<\eta_{2}<\eta_{m}}^{↑},{\left\{K\right\}}_{1^{-}}^{↑}$ of Kelvin principle. Within these models, these processes cannot be excluded by adding another process from a given model, because as one can check the operation of addition, it is an internal operation in these models. At the same time, the indicated two counter-examples are directly contrary to the CI principle.Now the errors in the supposed typical proof of K∣→CI implication will be pointed out. This typical pseudo-proof is proof by contradiction. It relies on the $↑0↑(-Q_{2},0,-Q_{2})$ process, which contradicts the CI principle ($Q_{2}>0$). This process is then combined with the engine process $↓\to ↓(Q_{1},W,Q_{2})$:
(33)$$\left(-Q_{2},0,-Q_{2}\right)+\left(Q_{1},W,Q_{2}\right)=\left(Q_{1}-Q_{2},W,0\right).$$ The resulting process ↓→ is a *perpetuum mobile* type II, which contradicts the *K* principle. Then it is claimed that the K∣→CI implication has been proved, but this is not true. First, it is not clear whether it is allowed to add a process ↓→↓ that complies with the CI principle, since we want to contradict this principle in the proof by contradiction. The result of adding this process is a process that formally no longer contradicts the CI principle. Secondly, the formal implication should be checked for all processes, not just for one type of processes ↑0↑ in the proof scheme by contradiction. There are more processes that contradict the CI principle, e.g., ↑→↑,0→↑.Thus, the structure of the *K* and CI models and the two counterexamples, as well as the alleged proof errors, disprove the implication K∣→CI. □

### 3.13. Theorem ¬(K∣→CII)

**Proof.** Based on (Equation 19) oraz (Equation 25) we can see that {K}⊄{CII}. The family of models for the *K* principle is too large to be a subset of the CII principle model. The *K* principle models include too efficient engine processes $\eta >\eta_{C}$, as well as refrigeration processes with too low efficiency $\tilde{\eta }<\eta_{C}$. Such processes, on the other hand, are contrary to the CII principle.The processes $↓\to ↓(Q_{1}-q,W,Q_{2}-q)$, $↑\leftarrow ↑(-Q_{1}-q,-W,-Q_{2}-q)$ are counterexamples for the implication under consideration. They were referenced to the $↓\to ↓(Q_{1},W,Q_{2})$ process with Carnot efficiency $\eta_{C}$. For q>0 these two counterexamples contradict the CII principle, since for them ΔS<0. However, they do not contradict the *K* principle, which does not allow 100% efficiency, but does not give a stronger limitation with Carnot efficiency.However, one might get the impression that the processes $\left(Q_{1}-q,W,Q_{2}-q\right)$ and $\left(-Q_{1}-q,-W,-Q_{2}-q\right)$ do not follow the *K* principle because, when added to the inverse $\left(-Q_{1},-W,-Q_{2}\right)$ or the regular $\left(Q_{1},W,Q_{2}\right)$ of the Carot process, they give respectively in both cases the process ↑0↑(−q,0,−q), which is impossible in the considered models (disregarding the existence of a separate class of models of the type ${\left\{K\right\}}^{↑}$). The point is, however, that in the model suitable for the $\left(Q_{1}-q,W,Q_{2}-q\right)$ process, there is no inverse Carnot cycle with Carnot efficiency ($\tilde{\eta }=\eta_{C}$). For the model suitable for the $\left(-Q_{1}-q,-W,-Q_{2}-q\right)$ process, there is no Carnot engine cycle with Carnot efficiency ($\eta =\eta_{C}$). Moreover, both counterexamples of processes belong to different models, so these processes should not be added, either.It is also worth pointing to the errors in the alleged proof by the contradiction of the implication under consideration. It is known that the ↑0↑(−q,0,−q) process for q>0 does not comply with the CII rule because in this case ΔS<0. By combining the above process with the ↓→↓(q+W,W,q) engine process, we get *perpetuum mobile* type II, i.e., the process ↓→0(W,W,0). However, this does not prove anything, since the proof of formal implication cannot be based on one example of a process impossible for a given principle (and consequently impossible for a second principle). Moreover, considering both the impossible and the possible process (for the CII principle) is unclear. However, consider another impossible process $↓\to ↓(Q_{1}-q,W,Q_{2}-q)$ with an efficiency greater than that of the Carnot process. Combining it with the Carnot inverse cycle $↑\leftarrow ↑(-Q_{1},-W,-Q_{2})$ we get the process ↑0↑(−q,0,−q) impossible for the CII principle, but not excluded by the *K* rule. Such a process ↑0↑ occurs, for example, in the model ${\left\{K\right\}}_{0}^{↕}$, which includes any refrigeration process ${\left\{K\right\}}_{0}^{↕}$, but does not contain any engine process ↓→↓.On the basis of the structure of the {K}, {CII} models, two counterexamples and pseudo-proof errors, the formal implication of K∣→CII has been rebutted. □

Three formal implications of the CII principle remain to be proved.

### 3.14. Theorem CII∣→C0

**Proof.** Based on (Equation 25) and (Equation 13) we can see that {CII}={C0}, so in particular also {CII}⊂{C0}. Thus, on the basis of the models (sets of processes), the considered implication holds (and even equivalence).However, we will show directly that the CII principle applied to the $↓\to ↓(q_{1},w,q_{2})$ engine process leads to the Carnot efficiency condition:
(34)$$\frac{-q_{1}}{T_{1}}+\frac{q_{2}}{T_{2}}\geq 0\;\;\;\Rightarrow \;\;\;\frac{-T_{2}}{T_{1}}\geq \frac{-q_{2}}{q_{1}}\;\;\;\Rightarrow \;\;\;1-\frac{T_{2}}{T_{1}}\geq 1-\frac{q_{2}}{q_{1}}\;\;\;\Rightarrow \;\;\;\eta_{C}\geq \eta .$$ Since the Carnot principle applies essentially to engine processes, the proof of formal implication may be limited here to such processes. On the other hand, the generality of the proof is additionally secured by the equality of the models. □

### 3.15. Theorem CII ∣→ *K*

**Proof.** Based on (Equation 25) and (Equation 19), (Equation 21) we can see that {CII}⊂{K}. There is even equality with the particular model of the result principle $\left\{CII\right\}={\left\{K\right\}}_{0<\eta_{m}<1}^{↓}$ for $\eta_{m}=\eta_{C}$.The analyzed implication is a weaker version of the implication (Equation 34):
(35)$$\eta \leq \eta_{C}=1-\frac{T_{1}}{T_{2}}<1.$$ Thus, the *K* principle is weaker than the CII principle. Since the *K* principle applies essentially to engine processes (or the absence of *perpetuum mobile* type II), the proof can be limited to processes of this type. On the other hand, the generality of the proof is secured by the fact that one model is included in the other model. □

### 3.16. Theorem CII ∣→ CI

**Proof.** Based on (Equation 25) and (Equation 15), (Equation 17) we can see that {CII}⊂{CI}. Strictly speaking, the first-principle model is equal to the special result-principle model $\left\{CII\right\}={\left\{CI\right\}}_{0<\eta_{m}<1}$ for $\eta_{m}=\eta_{C}$.However, we will show directly that the principle CII applied to the processes of heat flow (q,0,q) allows heat to flow downwards ↓0↓, but does not allow spontaneous flows upwards ↑0↑:
(36)$$\frac{-q}{T_{1}}+\frac{q}{T_{2}}\geq 0\;\;\;\Rightarrow \;\;\;\frac{T_{1}-T_{2}}{T_{1}T_{2}}q\geq 0\;\;\;\Rightarrow \;\;\;q\geq 0.$$ This condition expresses the essence of the CI principle (assuming $T_{1}>T_{2}$), so the proof need not include other processes. On the other hand, the generality of the formal implication is secured by the inclusion of the CII principle model in the CI principle model. □

Thus, six formal implications were proved and six other formal implications were rebutted. The proof was made double: the first method was based on the inclusion relationship of models, and the second method was direct proof (not by contradiction). The rebuttals, on the other hand, were threefold: the first method showed the absence of inclusion relationship of models, the second method was based on one or two counterexamples, and the third method indicated errors in the alleged pseudo-proof by contradiction.

## 4. Results

The proved and disproved implications of the four principles (C0,CI,K,CII) are summarized in Table 3. Thus, only the two principles C0 and CII turned out to be equivalent. These principles are also stronger than the CI and *K* principles that follow from the previous ones. Thus, what Smoluchowski postulated in 1914 is being implemented: *“We call the Carnot principle as the second law of thermodynamics since Clausius time”* [41,42]. It is worth noting that the equivalence of the principles C0 and CII occurs in the conceptual system of thermodynamics assuming convexity and completeness of models. These additional rules were needed to give general meaning to the principles C0, CI, and *K*. The convexity assumption in the sense of internality of addition is implicitly commonly used, and the completeness rule only extends this assumption. The CII principle, on the other hand, does not need these additional rules.

Another way to show relationships between principles is to represent relationships between sets of all their processes belonging to all models. It has been marked, in a simplified manner, in the diagram of Figure 5. We see that the largest set of possible processes is allowed by the Kelvin principle. The only process that is not covered by the Kelvin principle, and which the Clausius I principle does not exclude, is a *perpetuum mobile* type II. Both the Clausius I principle and the Kelvin principle allow for a *perpetuum mobile* type III and, thus, differ significantly from the principles of Carnot and Clausius II. The Kelvin principle additionally includes processes of spontaneous flow of heat upwards (towards a higher temperature).

The relationships between the principles are also shown in Figure 6 in a “logical square” design. One diagonal of the square indicates equivalence, and the other diagonal is not marked, as it is neither equivalence nor implication.

## 5. Conclusions

Of the four principles (Carnot, Clausius I, Kelvin, Clausius II) pretending to formulate the II law of thermodynamics, only the Carnot and Clausius II principles turned out to be equivalent and strong principles, that forbid the decrease of the entropy of a heat-insulated system. The principles of Clausius I and Kelvin turned out to be less restrictive principles that say little about the reversibility of processes, and even allow impossible processes. These principles are true, but they are also so obvious that their predictions contribute very little. Moreover, the principles of Clausius I and Kelvin turned out to be, strictly speaking, independent (they do not imply each other). However, if we omit the *perpetuum mobile* of the II kind, it can be said, that the Clausius I principle is stronger than the Kelvin principle.

The most elegant formulation of the II law of thermodynamics is the entropic principle of Clausius II. The only competing formulation of this law, as it turns out, is the equivalent Carnot principle. Carnot principle is often *implicite* treated as the equivalent of the II law of thermodynamics, but *explicite* was not included in the formulation of this principle. Instead, the formulations of the II law mistakenly included the Clausius I principle and the Kelvin principle (as well as the Caratheodory principle). It can be said that it has not been noticed that the Carnot condition $\eta \leq \eta_{C}$ is stronger than the Kelvin condition in terms of Ostwald η<1. The II law of thermodynamics excludes not only the *perpetuum mobile* type II, but also the *perpetuum mobile* type III, that is a heat engine with an efficiency greater, than that of the Carnot engine. In the context of Carnot, who was a historical pioneer of the II law of thermodynamics, the distractor was the lack of knowledge of the I law of thermodynamics in his time. Carnot used the concept of indestructible heat (*caloric*), but in the sense of heat, not energy. Nevertheless, it did not prevent him from formulating correct conclusions regarding the II law of thermodynamics.

So how to explain the incorrect proof of the equivalence of Clausius I and Kelvin principles with the II law of thermodynamics? The first cause and distractor may be the frequent consideration of only Clausius I and Kelvin principles, regardless of the entropy formulation (Clausius II), or even more so the Carnot principle. In this approach, we lose the stronger context of the II law of thermodynamics. This, however, does not explain the alleged proof of a substantially incorrect implication of K∣→CI. Here a second cause appears, based on the formal implication ∣→ distractor, which is a material implication ⇒. The formal implication requires more subtle methods of proving, than the material implication. The main difference is the necessity to use the universal quantifier in formal implication, i.e., the necessity to check all the processes belonging to the model of the principle, which is the antecedent of the implication.

## Figures and Tables

**Figure 1 entropy-24-00392-f001:**
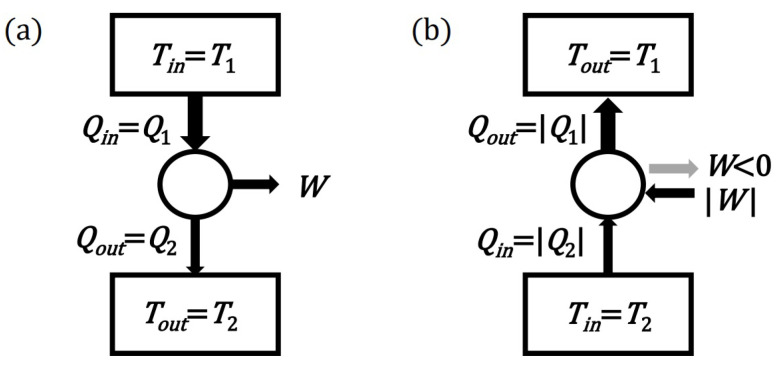
Schemes of two basic thermal devices: (**a**) A typical heat engine operating at the expense of the heat from the higher temperature reservoir. (**b**) A device that extracts heat from the lower temperature reservoir is necessarily (e.g., in the Carnot principle) a heat pump or a refrigerator that must be powered by work.

**Figure 2 entropy-24-00392-f002:**
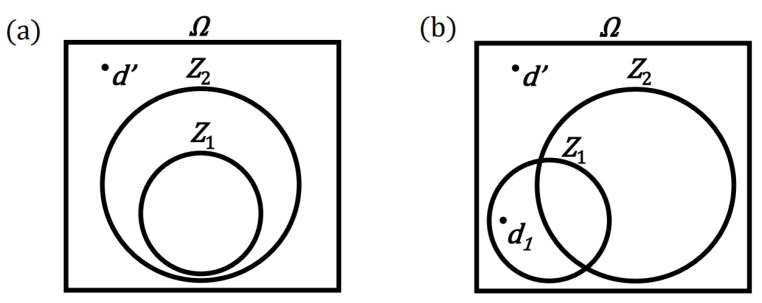
Schematic representation of the relationship between two physical principles $Z_{1}$ and $Z_{2}$ in a set of virtual physical processes Ω ({} brackets to distinguish the model from the principle are omitted here): (**a**) The case of implication of principles means the inclusion of appropriate sets (models). (**b**) A case of principles that do not implicate each other. The counterexample in the form of the $d_{1}$ process disproves the implication $Z_{1}|\;\to Z_{2}$, despite pointing to the $d^{\prime }$ process that would allegedly prove this implication by contradiction.

**Figure 3 entropy-24-00392-f003:**
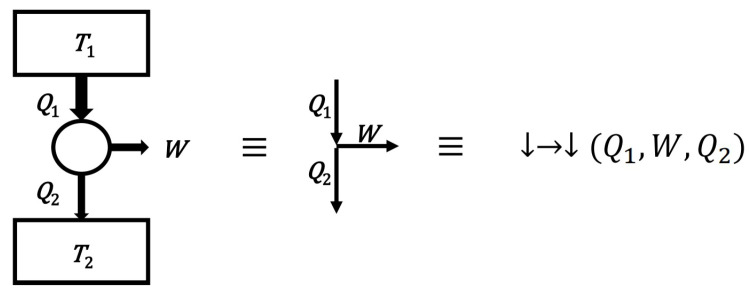
Schematic drawing of the engine process with the equivalent arrows diagram and the equivalent notation used in the text. Most often, the engine process denoted by capital letters will denote a Carnot efficient engine. The ↓→↓ engine process with a different efficiency may then be labeled as $\left(Q_{1}\pm q,W,Q_{2}\pm q\right)$ or $\left(q_{1},w,q_{2}\right)$.

**Figure 4 entropy-24-00392-f004:**
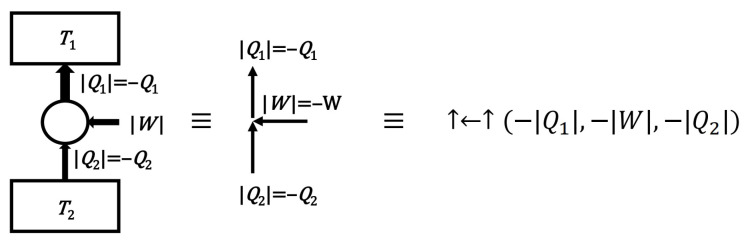
Schematic drawing of the cooler process with the equivalent arrows diagram and the equivalent notation used in the text. The directions of the arrows show the actual direction of heat flow or how the work was done. The heat and work symbols refer nominally to the engine process, so they have negative values here.

**Figure 5 entropy-24-00392-f005:**
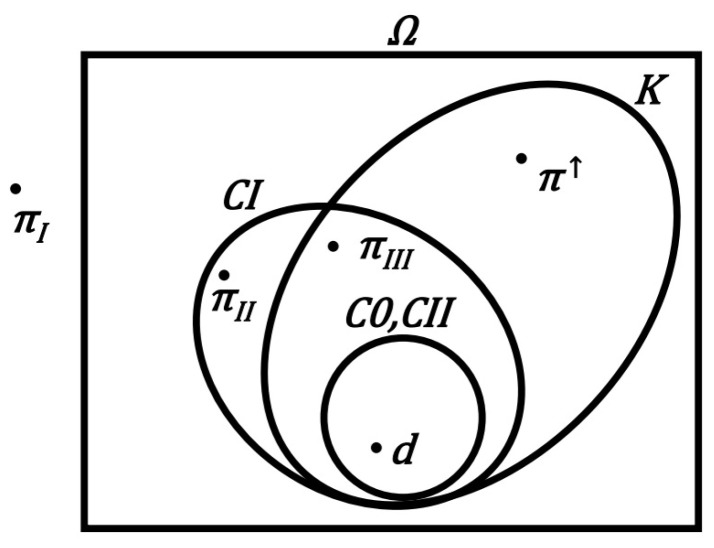
Resulting diagram showing the relationship between the fundamental principles crucial to the II law of thermodynamics. Additionally, the location of the fictitious *perpetuum mobile* processes of three types was marked, along with the heat flow process “upwards” $\pi^{↑}$. *Perpetuum mobile*$\pi_{I}$ goes beyond a dozen considerations, and it would be even more difficult to locate $\pi_{0}$ here. On the other hand, the physical process (diagram), consistent with the II law of thermodynamics (with principles C0 or CII), was denoted by *d*.

**Figure 6 entropy-24-00392-f006:**
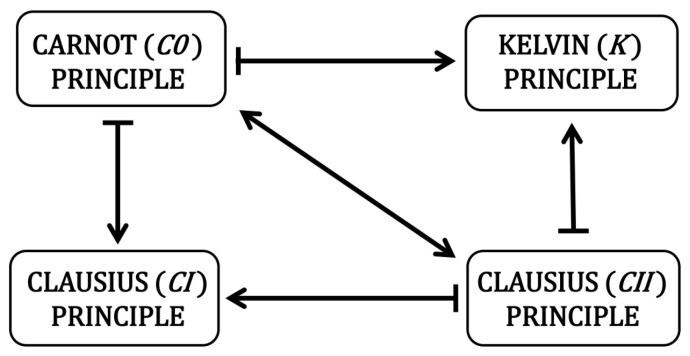
The resulting “logical square” of the implications of the four principles related to the II law of thermodynamics. This law is determined equivalently by the principles on the main diagonal. On the other hand, non-diagonal principles result from diagonal ones, but they are weaker and do not result from each other.

**Table 1 entropy-24-00392-t001:** Table of logical values of material implication and (material) equivalence.

*p*	*q*	p⇒q	p⇔q
0	0	1	1
0	1	1	0
1	0	0	0
1	1	1	1

**Table 2 entropy-24-00392-t002:** Full *status quo* matrix of the mutual implications of the four principles related to the II law of thermodynamics. Question mark “?” refers to the vague *status quo* of implication, and the annotation “(?)” refers to the implications questioned by the author.

Principle	Carnot	Clausius I	Kelvin	Clausius II
Carnot	≡	C0∣→CI ?	C0∣→K ?	C0∣→CII ?
Clausius I	CI∣→C0 (?)	≡	CI∣→K (?)	CI∣→CII (?)
Kelvin	K∣→C0 (?)	K∣→CI (?)	≡	K∣→CII (?)
Clausius II	CII∣→C0	CII∣→CI	CII∣→K	≡

**Table 3 entropy-24-00392-t003:** The resulting matrix of mutual formal implications (or the lack of them) of the four principles: Carnot, Clausius I, Kelvin, Clausius II. Each implication was proved or disproved, and was additionally analyzed on the basis of the structure of the models (sets of processes).

Principle	Carnot	Clausius I	Kelvin	Clausius II
Carnot	≡	C0∣→CI	C0∣→K	C0∣→CII
Clausius I	CI∣↛C0	≡	CI∣↛K	CI∣↛CII
Kelvin	K∣↛C0	K∣↛CI	≡	K∣↛CII
Clausius II	CII∣→C0	CII∣→CI	CII∣→K	≡

## Data Availability

Not applicable.

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
