# Peer review of "Proof of Equivalence of Carnot Principle to II Law of Thermodynamics and Non-Equivalence to Clausius I and Kelvin Principles"

_entropy, 2022, doi:10.3390/e24030392_

Round 1

Reviewer 1 Report

See attached file

Author Response

see PDF file

Reviewer 2 Report

In 1949, Max Born complained, “…classical thermodynamics proceeded in quite different way, introducing the conception of idealized thermal machines which transform heat into work and vice versa … The second law of thermodynamics is then derived from the assumption that not all processes of this kind are possible: you cannot transform heat completely into work, nor bring it from a state of lower temperature to one of higher ‘without compensation.’ These are new and strange conceptions, obviously borrowed from engineering … they deviated too much from the ordinary methods of physics …” (Max Born, Natural Philosophy of Cause and Chance [Oxford Univ. Press, 1949]). He went on to promote the mechanistic physics approach to the second law developed by Carathéodory. The classical formalism derived from what Born promoted has served a useful purpose for the application of thermodynamic theory but the Carathéodory formulation of the second law itself has never achieved the status that Born and Carathéodory intended of becoming the foundation of thermodynamic theory.

This paper by Koczan is a proof of the richness in the idealized thermal machines conception of the second law of thermodynamics. While I cannot judge the completeness of the logical argument of the paper, the paper has added considerable subtleness in the Carnot principle model (C0), models for Clausius I principle (CI), models for Kelvin principle (K), and the Clausius II principle model (CII)—and is spot on in its conclusion with regards to the four second-law principles that “the Kelvin principle and the Clausius I principle are not exhaustive formulations of the II law of thermodynamics.”

While I understand that in the context of Schemes of two basic thermal devices, the conclusion that C0 and CII are logically equivalent, it would be useful for the author to add in the Result or Conclusion of the paper a comment that CII, though logically equivalent to, is conceptually more general than C0 in that it is a principle directly applicable to all schemes of things.

With regards to the evaluation of the classical formalism and its introduction of quasi-static processes, I would like to add here the following statement I made in A Treatise of Heat and Energy (Springer, Dec. 7, 2019):

A reversible machine remains the best or natural approach to start the consideration of the concept of entropy, Eq. (62A).

   Once the introduction is made, classical formalism is correct in pointing out that reversibility is a too restrictive condition for defining entropy. Classical formalism is mistaken, however, in replacing reversibility with quasi-staticity. The modern formalism shows that quasi-staticity in the classical formalism, Eq. (83), is in fact internal reversibility, which is the necessary and sufficient condition for the definition of entropy, Eq. (92).

Author Response

see PDF file

Reviewer 3 Report

The author  investigates the equivalence relations among the multiple known statements of the second law of thermodynamics. He does that by introducing a mathematical formalism that is new to me. I would call it  the "vectorization of thermodynamic processes". In the intentions of the author,  it should help to prove which statement is stronger (higher in the inference chain, so to speak) and categorize them according to their strength (very useful is also the use of the Euler-Venn diagrams).

The paper is well written, the approach seems sound to me, and the results appear not to contradict what one would expect by intuition and reasoning. Therefore, I do not have any reasons not to recommend this paper for publication in this Journal.

Author Response

see PDF file
